# WEIGHT DECAY IMPROVES LANGUAGE MODEL PLASTICITY

## ABSTRACT

The prevailing paradigm in large language model (LLM) development is to pretrain a base model, then perform further training to improve performance and model behavior. However, hyperparameter optimization and scaling laws have been studied primarily from the perspective of the base model's validation loss, ignoring downstream adaptability. In this work, we study pretraining from the perspective of *model plasticity* – the base model's ability to adapt to downstream tasks through fine-tuning. We focus on the role of weight decay, a key pretraining hyperparameter. Through systematic experiments, we find that models pretrained with stronger weight decay are more plastic, showing larger performance gains when fine-tuned on downstream tasks. This phenomenon can lead to counter-intuitive trade-offs where base models that perform worse after pretraining can perform better after fine-tuning. Further investigation of weight decay's mechanistic effects on model behavior reveals that it encourages linearly separable representations, regularizes attention, and reduces overfitting on the training data. In conclusion, this work casts light on the multifaceted role of a single optimization hyperparameter in shaping model behavior and demonstrates the importance of using metrics beyond the cross-entropy loss for hyperparameter optimization.

## 1 INTRODUCTION

Weight decay is a canonical hyperparameter whose role has evolved from improving generalization in multi-epoch training to shaping optimization and training stability in large-scale single-epoch pretraining (Hardt et al., 2016; Zhang et al., 2017; Sun et al., 2025; D'Angelo et al., 2024; Zhang et al., 2025; Wang & Aitchison, 2024). Moreover, modern language models (LM) are developed in two stages – pretraining and post-training – but the stages are often decoupled, with hyperparameters optimized to minimize pretraining loss under the assumption this improves downstream performance (Hoffmann et al., 2022b; Bi et al., 2024). However, to what extent does this assumption hold?

We study the relationship between pretraining and post-training from the lens of *model plasticity* – a model's ability to adapt to new data upon further training (Berariu et al., 2021; Dohare et al., 2024) – which bridges the two stages. As we show, two models with similar pretraining loss can differ in plasticity, so optimizing for pretraining loss alone may not yield the best post-trained model.

We pretrain models with varying weight decay and evaluate their ability to learn during fine-tuning. Our experiments span two model families (Llama-2 and OLMo-2), multiple model sizes (up to 4B), compute-optimal and overtrained regimes, six Chain-of-Thought (CoT) tasks, and metrics for both solution correctness and quality. Our design aligns pretraining hyperparameter selection with the ultimate goal of maximizing downstream performance. Our contributions are as follows:

- We show that weight decay improves model plasticity and downstream performance. Across model families, scales, and training regimes, the evidence points to an optimal pretraining weight decay larger than the 0.1 default. This calls for potentially re-evaluating standard hyperparameter choices to better account for a model's downstream adaptability.

- We provide one of the first examples where optimizing hyperparameters to minimize pretraining loss does not yield the best downstream performance. We find a regime where larger weight decay leads to higher pretraining loss *and* better downstream performance.

- We study the mechanisms through which weight decay shapes model behavior, finding it encourages linearly separable representations, regularizes attention, and reduces overfitting. These effects may explain how weight decay promotes plasticity.

## 2 Related Work

**Weight decay in language model training.** Beyond its classical role in regularization and generalization (Krogh & Hertz, 1991; Zhang et al., 2018; Zhou et al., 2024), weight decay plays other roles in LM training, such as improving training stability and optimization and inducing low-rank attention (D'Angelo et al., 2024; Li et al., 2020; Kosson et al., 2024; 2025; Wen et al., 2025; Kobayashi et al., 2024). While prior work studied how to set weight decay to minimize pretraining loss (Bergsma et al., 2025; Kim et al., 2025), this paper studies its impact on model plasticity.

**Model plasticity.** Model plasticity has been studied in continual, transfer, and reinforcement learning, where models undergo multiple training rounds (Dohare et al., 2024; Klein et al., 2024; Coetzer et al., 2025). Prior works have developed approaches to improve plasticity (Ash & Adams, 2020; Dohare et al., 2024; Kumar et al., 2023; Miconi et al., 2018). In constrast to prior work on forgetting and tokenization in LM plasticity (Chen et al., 2023; Abagyan et al., 2025), this paper investigates the role of weight decay, a standard hyperparameter. Additional details are in Appendix B.1.

## 3 Background and Methods

**Weight decay in AdamW.** This paper focuses on the weight decay hyperparameter $\lambda$ in AdamW (Loshchilov & Hutter, 2019). In LM pretraining, the choice $\lambda = 0.1$ has emerged as a kind of default (Brown et al., 2020a; Touvron et al., 2023; OLMo et al., 2024).

**Language model plasticity.** Following prior work (Berariu et al., 2021; Dohare et al., 2024), we measure plasticity by fine-tuning the model and evaluating performance on the fine-tuning task: the better the performance, the better the model learned during fine-tuning, thus the higher the plasticity.

> **Research Question.** How does weight decay during pretraining affect model plasticity, i.e. the pretrained model's ability to learn new knowledge during subsequent training?

To study this, we pretrain five Llama-2 and OLMo-2 models with varying weight decay (Llama-2-0.5B-20x, Llama-2-1B-20x, Llama-2-4B-20x, OLMo-2-1B-20x, and OLMo-2-1B-140x). These models span different sizes (up to 4B) and training regimes (Chinchilla-optimal and overtrained). We fine-tune (SFT) these models on six CoT tasks – MetaMathQA, MedMCQA, PubMedQA, MMLUProCoT, RACE, and SimpleScaling (Yu et al., 2023; Pal et al., 2022; Jin et al., 2019; Wang et al., 2024; Lai et al., 2017; Muennighoff et al., 2025) – and evaluate them using six metrics: Greedy, Maj@16, RM@16, Pass@16, Correct Ratio, and ORM Score. More details in Appendix B.2.

## 4 Weight decay Improves Language Model Plasticity

We present experimental results. We identify the optimal weight decay based on pretraining performance (Section 4.1) and based on model plasticity and downstream performance (Section 4.2), and examine the relationship between pretraining and downstream performance (Section 4.3).

### 4.1 The optimal pretraining weight decay based on pretraining performance

We identify the weight decay value that minimizes cross-entropy validation loss after pretraining, i.e., the value considered optimal by current approaches (Bergsma et al., 2025). We pretrain models with varying weight decay and plot their validation cross-entropy loss in Figure 1. At 20 TPP, for both Llama-2 and OLMo-2 models, the optimal weight decay is larger than the default of 0.1. At 140 TPP, for the OLMo-2-1B-140x model, the default value of 0.1 outperforms (leads to a lower validation loss than) larger values of 0.3 and 1.0. These results are consistent with previous analyses which recommend decreasing weight decay as training time (TPP) increases (Bergsma et al., 2025).

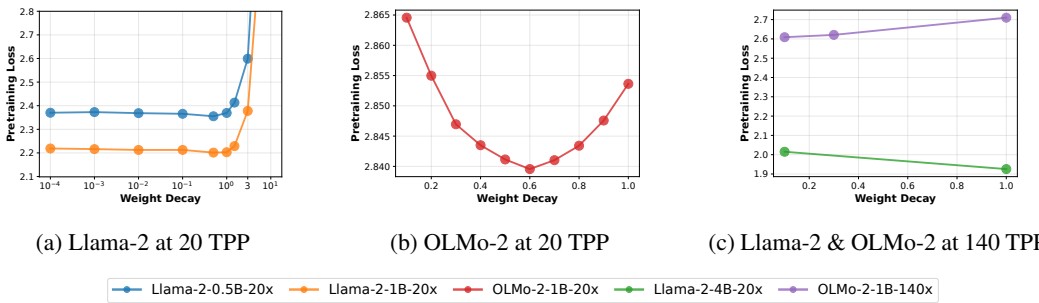

(a) Llama-2 at 20 TPP    (b) OLMo-2 at 20 TPP    (c) Llama-2 & OLMo-2 at 140 TPP

Figure 1: **Pretraining validation cross-entropy loss of models pretrained with varying weight decay.** The weight decay that minimizes pretraining loss may equal exceed the 0.1 default.

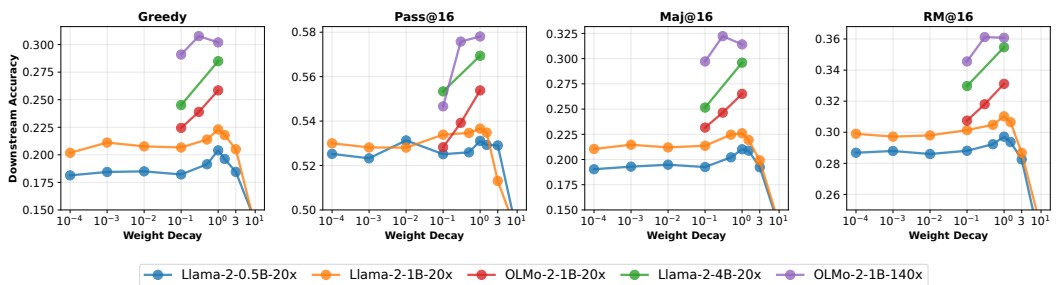

Figure 2: **Weight decay during pretraining improves language model plasticity and downstream performance.** The optimal weight decay for downstream performance is larger than the standard default of 0.1. In addition, the optimal weight decay based on pretraining performance (Figure 1) and that based on downstream performance (this figure) are different.

## 4.2 THE OPTIMAL PRETRAINING WEIGHT DECAY BASED ON DOWNSTREAM PERFORMANCE

Next, we fine-tune the pretrained models from Section 4.1 on six tasks and evaluate their performance on these tasks. Figure 2 shows the models' average downstream accuracy across all tasks (detailed results in Appendix E). Among models with reasonable pretraining losses (i.e., candidates for subsequent training), higher pretraining weight decay leads to greater model plasticity, enabling the pretrained model to perform better on the fine-tuning task. This finding is consistent Across model families, sizes, training regimes, fine-tuning tasks, and metrics, models pretrained with weight decay higher than the 0.1 default perform better downstream. At 20 TPP, the optimal pretraining weight decay is 1.0 (Llama-2-{0.5B and 1B}-20x, Llama-2-4B-20x, and OLMo-2-1B-20x). At 140 TPP, it is 0.3 (OLMo-2-1B-140x). In addition, the weight decay that minimizes pretraining loss (Figure 1) and the one that maximizes downstream accuracy (Figure 2) are different for each model.

> **Finding 1.** Pretraining weight decay can improve model plasticity and downstream performance. The optimal pretraining weight decay for plasticity is larger than the 0.1 default.

## 4.3 THE RELATIONSHIP BETWEEN PRETRAINING LOSS AND DOWNSTREAM PERFORMANCE

Next, we examine the relationship between a model's pretraining validation cross-entropy loss and its accuracy on tasks after fine-tuning, plotted in Figure 3. Although their Pearson correlation ($r$) is negative for 20 TPP models and positive for 140 TPP models, this relationship is not stable (Figure 7). Examining pairs of points, we see that models with better pretraining performance can perform better or worse downstream, and models with similar pretraining performance can perform differently downstream. For example, OLMo-2-1B-140x pretrained with weight decay 0.3 or 1.0 performs slightly worse after pretraining (pretraining validation loss: 2.6208 and 2.7064, respectively) than when pretrained with weight decay 0.1 (pretraining validation loss: 2.6088), but the former two pretrained models perform noticeably better after fine-tuning (Figure 2).

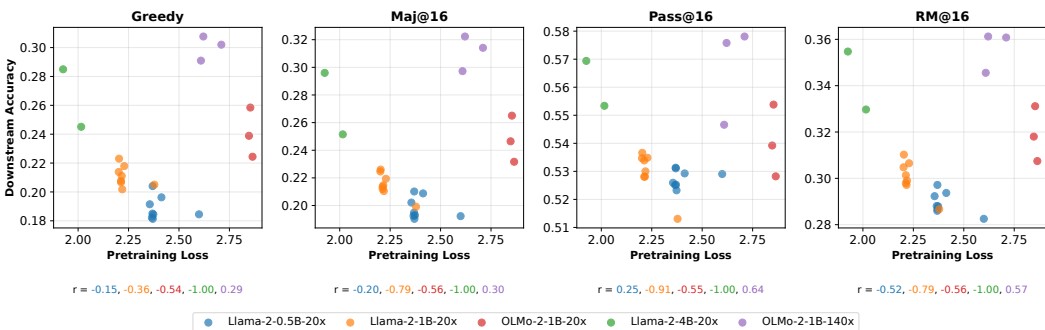

Figure 3: **Pretraining performance is not fully predictive of downstream performance.** Models with lower pretraining cross-entropy validation loss can perform better or worse after fine-tuning than models with higher pretraining losses.

> **Finding 2.** The pretraining weight decay value that minimizes the cross-entropy validation loss does not necessarily lead to the best downstream performance.

## 5 WEIGHT DECAY AND MODEL BEHAVIOR: A MECHANISTIC PERSPECTIVE

We now explore how weight decay may improve plasticity. Detailed results are in Appendix B.4.

**Weight decay encourages linearly separated representations.** Our linear probing experiments show that models pretrained with higher weight decay have more structured representations (Figure 4), potentially enabling fine-tuning to start from better features and achieve higher performance. This aligns with previous findings (Lee et al., 2023) and supported by further analyses (Figure 15).

**Weight decay reduces the rank of attention matrices.** We measure the pseudo-rank (Appendix F.2.1) of attention matrices during training and find that larger weight decay reduces their rank (Figure 5). Thus, weight decay may promote the learning of higher-level patterns that are more useful during fine-tuning, resulting in higher downstream performance.

**Weight decay reduces overfitting on training data.** We examine the difference between the validation loss and training loss (i.e., the fully trained model's average loss on the training data). We find that weight decay decreases this difference, indicating that it reduces overfitting (Figure 6).

> **Finding 3.** Pretraining weight decay has diverse mechanistic effects on model behavior. It encourages linearly separated representations, regularizes attention, and reduces overfitting.

## 6 DISCUSSION AND CONCLUDING REMARKS

This work provides a multidimensional characterization of weight decay in the LM training. We show that lower weight decay improves pretraining loss, while higher weight decay improves plasticity and downstream performance (Section 4). Weight decay's role in plasticity might be explained through various mechanisms (Section 5). These findings provide one of the first demonstrations that optimizing hyperparameters based solely on pretraining performance may not yield the best downstream model. This means that plasticity gains must be weighed against other training parameters. For example, under heavy overtraining (Comanici et al., 2025), substantially lower pretraining loss may outweigh the benefits of plasticity. In addition, the optimal weight decay can vary based on the objective – from plasticity to training dynamics (Hoffmann et al., 2022b; D'Angelo et al., 2024; Kosson et al., 2025) – requiring weighing trade-offs and prioritizing objectives. Future work may investigate the stability-plasticity trade-off in more detail and weight decay's role in plasticity for foundation models beyond language (e.g., multimodal models) and for other downstream desiderata (e.g., safety alignment).

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

APPENDIX

***Appendix A answers questions for the workshop challenge submission***.

Appendix B provides a fuller discussion of sections of the main paper.

Appendix C provides details on model pretraining.

Appendix D provides details on model fine-tuning and datasets.

Appendix E provides additional experimental results for Section 4.

Appendix F provides additional results for Section 5.

## A  SCIENCE OF DL IMPROVEMENT CHALLENGE SUBMISSION

### A.1  WHAT MODEL ARE YOU TARGETING?

*Provide a summary of the problem the deep net model is designed to solve. Good summaries should outline the state of the literature, provide an overview that domain experts would consider reasonable, and cite relevant sources.*

**Research question and target model.** This paper studies how weight decay shapes the plasticity of language models.

**Investigating weight decay's new role in model plasticity.** Weight decay is a canonical hyperparameter in deep learning whose role has evolved alongside changes in training regimes. In classical multi-epoch training, weight decay was understood primarily as a regularizer that improves generalization by shrinking weights and controlling model capacity (Hardt et al., 2016; Zhang et al., 2017; Sun et al., 2025). In contemporary large-scale pretraining, which often uses a single epoch over massive datasets (Brown et al., 2020b; Kaplan et al., 2020), weight decay no longer primarily serves the purpose of generalization but plays other roles, such as improving optimization and training stability (D'Angelo et al., 2024; Zhang et al., 2025; Wang & Aitchison, 2024; Li et al., 2020; Kosson et al., 2024; 2025). This paper studies weight decay's role in language model plasticity, a new role for this canonical hyperparameter that is critical to how modern language models are trained.

**The benefit of the model plasticity perspective for language models.** Model plasticity is the ability of a trained model to effectively adapt to new data upon further training, modifying its parameters and internal representations in response to the new data and enabling effective learning of new tasks without reinitialization (Berariu et al., 2021; Dohare et al., 2024). Model plasticity naturally bridges pretraining and post-training, the two main training stages for modern language models, capturing how readily a pretrained model (base model) can be reshaped for downstream tasks. This is in contrast to current practices which often treat the two functionally-linked stages as decoupled, studying pretraining hyperparameters and scaling laws predominantly through the lens of the pretrained model's validation loss, assuming that a lower pretraining validation loss also yields a more capable downstream model (Hoffmann et al., 2022b; Bi et al., 2024). The model plasticity perspective in this paper aligns pretraining hyperparameter selection with the ultimate objective of maximizing performance after further training.

### A.2  HOW DO YOUR RESULTS CONTRIBUTE—OR COULD POTENTIALLY CONTRIBUTE—TO UNDERSTANDING THESE MODELS?

*What aspects of the models become better understood thanks to your work?*

This paper improves our understanding of language models in the following ways.

**We find that weight decay during pretraining improves the plasticity of pretrained models.** Language models pretrained with stronger weight decay have greater plasticity and can better adapt to new data during fine-tuning, leading to final models with better performance.

**We find counterintuitive trade-offs between pretraining and downstream performance.** In contrast to prior work which optimizes pretraining hyperparameters for pretraining performance (Hoffmann et al., 2022b; Bi et al., 2024), this paper optimizes pretraining hyperparameters for the

ultimate goal of high downstream performance. In doing so, we find counterintuitive trade-offs: models that perform worse during pretraining can have greater plasticity (i.e., a better ability to adapt to new data) and perform better after fine-tuning. Thus, optimizing pretraining hyperparameters for pretraining performance alone may not always yield the best final model.

**We find that weight decay shapes the behavior of the pretrained model by encouraging linearly separable internal representations, regularizing attention matrices, and reducing overfitting on the training data.** Through these mechanisms, weight decay may promote the learning of higher-level patterns (rather than specific details or noise) during pretraining, producing a pretrained model with more useful information for fine-tuning and leading to better performance after fine-tuning. In this way, these mechanisms may explain why weight decay during pretraining improves language model plasticity.

### A.3 How do you expect your submission to influence future work?

*Propose ways in which your insights, findings, or methodologies could shape subsequent research directions, model design choices, or scientific applications.*

**Our findings motivate future work to further explore the relationship between weight decay and model plasticity.** Our findings highlight the multifaceted role of weight decay in language model behavior. While weight decay has previously been shown to improve optimization and training stability (D'Angelo et al., 2024), shape the learning rate (Li et al., 2020; Kosson et al., 2024; 2025), and control the effective step size (Wen et al., 2025) in language models, our findings reveal yet a new role for decay: it improves language model plasticity through various potential mechanisms (improving internal model representations, regularizing attention, and reducing overfitting). Future work may explore how weight decay shapes plasticity for foundation models beyond language (e.g., multimodal foundation models), for post-training methods beyond supervised fine-tuning (e.g., RL-based methods), and for other downstream desiderata (e.g., safety alignment).

**Our findings show the complexity of pretraining hyperparameter optimization, motivating future work to examine the downstream impact of pretraining hyperparameters.** First, we find that the optimal pretraining weight decay for model plasticity is likely higher than the default weight decay value of 0.1, calling for future work to re-consider standard hyperparameter choices. Second, we find that optimizing pretraining hyperparameters based on pretraining validation loss alone is not guaranteed to produce the best final model, i.e., models that perform worse after pretraining can perform better after fine-tuning. This counterintuitive trade-off calls for re-evaluating current practices and fundamental assumptions in hyperparameter optimization.

## B More Detailed Discussion of Sections in the Main Paper

### B.1 Related Work

**Weight decay in language model training.** Weight decay is a standard hyperparameter in language model training and is commonly implemented in conjunction with adaptive optimizers such as AdamW Loshchilov & Hutter (2017); Brown et al. (2020a); Grattafiori et al. (2024); OLMo et al. (2024); Liu et al. (2024). Beyond its classical role in regularization and generalization Krogh & Hertz (1991); Zhang et al. (2018); Loshchilov & Hutter (2017); Zhou et al. (2024), weight decay has also been shown to play other roles in language model training, such as improving optimization and training stability D'Angelo et al. (2024), shaping the learning rate Li et al. (2020); Kosson et al. (2024; 2025), controlling the effective step size Wen et al. (2025), inducing low-rank attention layers Kobayashi et al. (2024), and increasing forgetting of contaminated benchmark questions Bordt et al. (2025). Wang & Aitchison (2024) show that the weights of AdamW can be understood as an exponential moving average, and that the weight decay parameter plays a critical role in controlling its time scale. Bergsma et al. (2025) study how to set weight decay to minimize the pretraining loss of language models, finding that lower weight decay improves pretraining loss in the overtrained (high TPP ratio) regime. Kim et al. (2025) show that larger weight decay improves pretraining loss in the multi-epoch setting. In contrast to previous work which primarily focuses on weight decay's effects on the pretrained model, this paper examines how weight decay during pretraining shapes model plasticity.

**Plasticity of deep learning models.** Model plasticity has previously been studied in the contexts of continual learning, transfer learning, and reinforcement learning, settings in which models often undergo multiple rounds of training Dohare et al. (2024); Klein et al. (2024); Coetzer et al. (2025). Prior works have demonstrated that image models lose plasticity when subjected to additional rounds of training on new data, leading to a decreased ability to learn this new data Dohare et al. (2024); Lyle et al. (2023); Klein et al. (2024). Various approaches have been developed to improve model plasticity, including shrinking and perturbing model weights at the start of each training round Ash & Adams (2020), identifying and re-initializing less-useful model weights during training Dohare et al. (2024), pushing weights towards initialization weights during training Kumar et al. (2023), and learning per-connection plasticity strengths among neuron pairs Miconi et al. (2018). While previous studies have examined how active forgetting and tokenization Chen et al. (2023); Abagyan et al. (2025) affect language model plasticity, research on language model plasticity remains under-developed. In contrast to these works, this paper investigates the role of weight decay, a standard hyperparameter for language model training, on language model plasticity.

## B.2 BACKGROUND AND METHODS

**Weight decay in AdamW.** Motivated by prior findings that regularization helps vision models maintain plasticity Dohare et al. (2024), this paper investigates weight decay's role in language model plasticity. We focus on the weight decay hyperparameter $\lambda$ in the AdamW optimizer which, for each optimizer step $t \geq 1$, performs two decoupled updates: a gradient update given by

$$\theta_t = \hat{\theta}_t - \gamma_t \, \hat{m}_t / (\sqrt{\hat{v}_t} + \epsilon) \tag{1}$$

followed by a weight decay update given by

$$\hat{\theta}_t = \theta_{t-1} - \gamma_t \lambda \theta_{t-1} \tag{2}$$

based on model parameters $\theta_t$, learning rate $\gamma_t$, and first- and second-order moment estimates of the gradient $\hat{m}_t$ and $\hat{v}_t$ Loshchilov & Hutter (2019). For language model pretraining, the choice $\lambda = 0.1$ has emerged as a kind of default, used in many pretraining runs where the optimization hyperparameters are known Brown et al. (2020a); Touvron et al. (2023); OLMo et al. (2024).

**Language model plasticity.** To assess the plasticity of a pretrained model, we fine-tune the model on a task and then measure its performance on this task. The better the performance on this downstream task, the better the pretrained model was able to learn new data during fine-tuning, thus the higher the plasticity of the pretrained model. This approach to measuring model plasticity is consistent with prior literature Berariu et al. (2021); Dohare et al. (2024).

In this context, we now specify the research question:

> **Research Question.** How does weight decay during language model pretraining affect model plasticity, i.e. the pretrained model's ability to learn new knowledge during subsequent training?

We investigate this research question empirically. We perform experiments that systematically vary weight decay during pretraining, then fine-tune and evaluate the models' performance on various downstream tasks. Our experiments span various model families, model sizes, training regimes (TPP ratios), and fine-tuning tasks. The setup is as follows.

**Pretraining.** We train Llama-2 models on the FineWeb-Edu dataset Penedo et al. (2024) and OLMo-2 models on the OLMo-Mix-1124 dataset. We vary model size and TPP ratio, training models at the 20 TPP Chinchilla-optimal ratio Hoffmann et al. (2022a) and at the 140 TPP overtrained ratio. This setup yields five models: Llama-2-0.5B-20x, Llama-2-1B-20x, Llama-2-4B-20x, OLMo-2-1B-20x, and OLMo-2-1B-140x. For each model, we pretrain variants with different weight decay.

**Fine-tuning.** We perform supervised fine-tuning (SFT) of the pretrained models across six CoT tasks spanning various domains: MetaMathQA (math reasoning), MedMCQA (medical reasoning), PubMedQA (biomedical research), MMLUProCoT (general knowledge and reasoning), RACE (reading comprehension), and SimpleScaling (math, science, and logical reasoning) Yu et al. (2023); Pal et al. (2022); Jin et al. (2019); Wang et al. (2024); Lai et al. (2017); Muennighoff et al. (2025).

**Evaluation of model performance after fine-tuning.** We evaluate the fine-tuned models in a zero-shot manner, prompting them to generate solutions to questions in the fine-tuning test sets, and assess both the correctness and quality of the solutions using six evaluation metrics.

- **Greedy** (i.e., **Pass@1**): A single deterministic response is generated (temperature = 0). The question is marked correct if this response is correct.
- **Maj@16**, **RM@16**, and **Pass@16**: Sixteen responses are sampled (temperature = 1). The question is marked correct if the majority answer is correct (Maj@16), if the response with the highest ORM score is correct (RM@16), or if any of the responses are correct (Pass@16).
- **Correct Ratio**: Sixteen responses are sampled (temperature = 1). Among questions with at least one correct response, we compute the proportion of correct responses out of the sixteen sampled responses.
- **ORM Score**: In addition solution correctness, we also measure solution quality. Sixteen responses are sampled (temperature = 1). Each response is assigned a score using an outcome reward model (Skywork-Reward-Llama-3.1-8B-v0.2; ORM) and the average ORM score is computed.

Since these experiments span pretraining and fine-tuning, we adopt the end-to-end analysis framework from Qi et al. (2025). Additional setup details are in Appendix C and D.

## B.3 WEIGHT DECAY IMPROVES LANGUAGE MODEL PLASTICITY

We present the main experimental results. We begin by identifying the optimal pretraining weight decay based on pretraining performance (Section B.3.1), a common way to select pretraining hyperparameters Hoffmann et al. (2022a). Next, we investigate how weight decay shapes the plasticity of the pretrained model and identify its optimal pretraining value based on downstream performance (Section B.3.2). Then, we examine whether a model's pretraining performance is fully predictive its downstream performance (Section B.3.3).

### B.3.1 THE OPTIMAL PRETRAINING WEIGHT DECAY BASED ON PRETRAINING VALIDATION LOSS

We first identify the weight decay value that leads to the lowest cross-entropy validation loss after pretraining. This is the value considered optimal by current approaches in hyperparameter optimization for LLM pretraining (Bergsma et al., 2025). Following Bergsma et al. (2025), we pretrain various models by sweeping over different weight decay values and fixing all other hyperparameters. Figure 1 shows the validation cross-entropy loss of these pretrained models.

Extremely small weight decay values during pretraining do not have a significant effect on pretraining loss (Figure1a). On the other hand, extremely large weight decay values can result in very high pretraining loss, significantly degrading pretraining performance (Figure1a). At 20 TPP, for both Llama-2 and OLMo-2 models, we find that the optimal weight decay parameter is larger than the default of 0.1. In particular, among the weight decay values examined, the optimal weight decay is 0.5 for Llama-2-{0.5B and 1B}-20x, 0.6 for OLMo-2-1B-20x, and 1.0 for Llama-2-4B-20x. However, this relationship changes as training time increases. At 140 TPP, for the OLMo-2-1B-140x model, the default value of 0.1 outperforms (leads to a lower validation loss than) larger values of 0.3 and 1.0. This result that overtrained models have a lower optimal weight decay is consistent with previous analyses on weight decay scaling laws which recommend decreasing the value of the weight decay hyperparameter as training time (TPP) increases to optimize for pretraining validation loss Bergsma et al. (2025).

### B.3.2 THE OPTIMAL PRETRAINING WEIGHT DECAY BASED ON DOWNSTREAM PERFORMANCE

Next, we investigate how weight decay during pretraining affects model plasticity and downstream model performance. We fine-tune the pretrained models from Section B.3.1 (which were trained with varying weight decay) on six CoT tasks and evaluate the final models' performance on these tasks. Figure 2 shows the average downstream accuracy of the models across the six tasks based on four metrics. The downstream accuracy for individual tasks and based on all six metrics is in Appendix E.

Among models that achieved reasonable pretraining validation losses in Section B.3.1 (i.e., models that are suitable candidates for subsequent training), higher weight decay during pretraining confers a higher degree of model plasticity, enabling the pretrained model to learn better during fine-tuning and perform better on the fine-tuning task. The results show that models pretrained with weight decay higher than the default 0.1 value perform better on downstream tasks. This finding is consistent across model families (Llama-2 and OLMo-2), model sizes (up to 4B parameters), training regimes (20 TPP and 140 TPP), fine-tuning tasks (six tasks spanning various domains), and evaluation metrics (six metrics measuring both solution correctness and quality). Among the weight decay values examined, in the compute-optimal 20 TPP regime, the optimal pretraining weight decay is 1.0 (Llama-2-0.5B-20x, Llama-2-1B-20x, Llama-2-4B-20x, and OLMo-2-1B-20x). In the overtrained 140 TPP regime, the optimal pretraining weight decay is 0.3 (OLMo-2-1B-140x). It is possible that as models are trained for even longer (i.e., beyond 140 TPP), the optimal pretraining weight decay that leads to best downstream model performance may continue to decrease (this is an extrapolation that would need to be validated).

We also compare the weight decay value that minimizes pretraining validation loss (Figure 1 in Section B.3.1) with the value that maximizes task accuracy after fine-tuning (Figure 2 in this section). We find that these two weight decay values differ for each model. This shows that the "optimal" weight decay during pretraining is not absolute – it depends on the intended objective, such as optimizing for pretraining performance or downstream performance.

> **Finding 1.** Pretraining weight decay can improve model plasticity and lead to better downstream performance. The optimal pretraining weight decay value for plasticity is larger than the default of 0.1.

### B.3.3 THE RELATIONSHIP BETWEEN PRETRAINING LOSS AND DOWNSTREAM PERFORMANCE

Following the findings from the previous sections, we now investigate whether a model's pretraining performance is predictive of its downstream performance. We examine the pretraining validation cross-entropy loss of the pretrained models from Section B.3.1 and their accuracy on tasks after fine-tuning (measured in Section B.3.2). The relationship between these two variables is plotted in Figure 3.

We compare models with the same training setup (same model family, size, and TPP) that differ only in the pretraining weight decay hyperparameter. Although the Pearson correlation coefficient $r$ between pretraining and downstream performance is negative for models trained at 20 TPP (Llama-2-0.5, Llama-2-1B, Llama-2-4B, and OLMo-2-1B) and positive for models trained at 140 TPP (OLMo-2-1B), this correlation should be interpreted cautiously because the relationship is not visually apparent. In addition, re-computing the correlation coefficient by removing one observation at a time can change the magnitude and even the sign of the coefficient, suggesting this relationship is not very stable (Appendix Figure 7). By examining pairs of points, the results show that models with better pretraining performance (lower loss after pretraining) can perform better downstream (such observations exist for all five models) or worse downstream (such observations exist for Llama-2-0.5B-20x, Llama-2-1B-20x, and OLMo-2-1B-140x). In addition, models with similar pretraining performance can perform differently downstream (such observations exist for Llama-2-0.5B-20x, Llama-2-1B-20x, and OLMo-2-1B-20x). For example, OLMo-2-1B-140x pretrained with weight decay 0.3 or 1.0 performs slightly worse after pretraining (achieving pretraining cross-entropy validation losses of 2.6208 and 2.7064, respectively) than the same model pretrained with weight decay 0.1 (which achieves a pretraining cross-entropy validation loss of 2.6088), but the former two pretrained models perform noticeably better after fine-tuning (Figure 2, purple line). Altogether, these results indicate that pretraining performance is not necessarily predictive of downstream performance.

> **Finding 2.** The pretraining weight decay value that minimizes the cross-entropy validation loss does not necessarily lead to the best downstream performance.

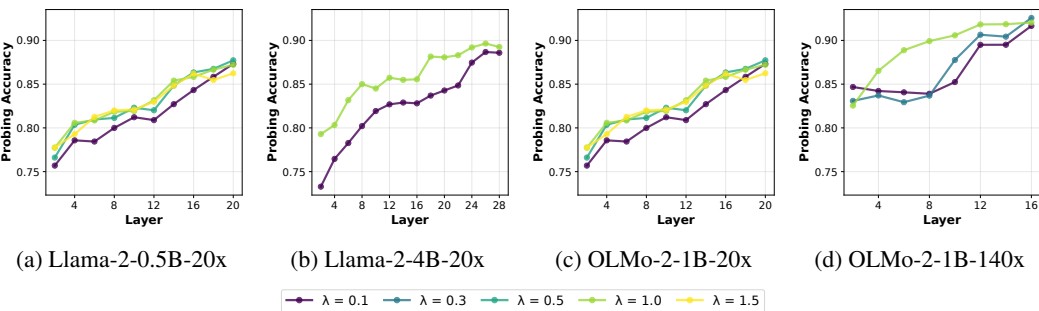

(a) Llama-2-0.5B-20x  (b) Llama-2-4B-20x  (c) OLMo-2-1B-20x  (d) OLMo-2-1B-140x

Figure 4: **Weight decay encourages linearly separated representations.** This figure depicts the accuracy of linear probes for sentiment and topic for models pretrained with different weight decay values. We observe that linear probing achieves better accuracy when models are pretrained with a weight decay greater than the default 0.1.

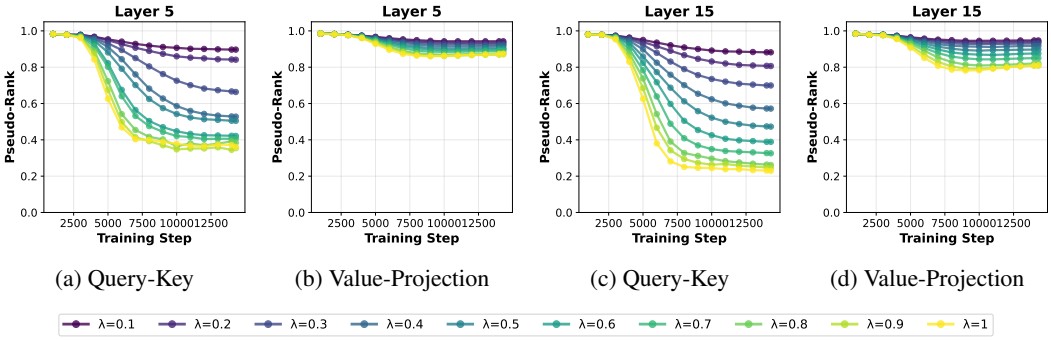

(a) Query-Key  (b) Value-Projection  (c) Query-Key  (d) Value-Projection

Figure 5: **Weight decay reduces the rank of attention matrices.** This figure depicts the average pseudo-rank (Appendix F.2.1) of the query-key ($W_{QK}$) and value projection ($W_{VP}$) matrices in layers 5 and 15 during the training of OLMo-2-1B models at 20 TPP.

### B.4 A MECHANISTIC PERSPECTIVE ON WEIGHT DECAY AND MODEL BEHAVIOR WORK

Prior work has shown that various factors can influence model plasticity, including the initialization state of model weights at the start of subsequent training, data representation (e.g., tokenization and categorical output representations), and model architecture (e.g., normalization layers) Ash & Adams (2020); Abagyan et al. (2025); Lyle et al. (2023). In Section B.3, we find that weight decay also shapes model plasticity. In this section, we explore three mechanisms through which weight decay shapes model behavior: how weight decay shapes the pretrained model's internal representations, attention matrices, and the extent to which it overfits the pretraining data. We also discuss how each mechanism might explain why weight decay improves language model plasticity.

#### B.4.1 WEIGHT DECAY ENCOURAGES LINEARLY SEPARATED REPRESENTATIONS

Inspired by previous findings that weight decay leads to more structured representations in vision models Jacot et al. (2024), we investigate the effect of weight decay on the representations learned by pretrained language models. We pretrain models with varying weight decay, obtain the last-token embeddings for different types of text at a given model layer, and train a linear probe to classify these embeddings. We examine two tasks, classifying text based on sentiment (positive or negative movie reviews from the Stanford Sentiment Treebank dataset; Socher et al. (2013)) or topic (four types of news articles from the AG News dataset; Zhang et al. (2015)). The average accuracy of these linear probes over the two tasks is shown in Figure 4 (accuracy for individual tasks are in Appendix F.1).

We observe that when a given model is pretrained with higher weight decay, the accuracy of the linear probe trained on the model's representations tends to be higher at every layer of the model. While this relationship is not perfectly monotonic (in some instances, a slightly higher weight decay

can lead to a similar or slightly lower probing accuracy), it is generally consistent across weight decay values and model layers. In addition, we observe this relationship across model families, sizes, and training regimes (i.e., for all five model setups). Thus, through these linear probing experiments, we find that representations from models pretrained with higher weight decay result in higher probing accuracies, indicating that they are more linearly separated and suggesting that these models form more structured representations.

The finding that weight decay shapes the representations of pretrained language models points to a potential explanation for why weight decay improves model plasticity (Section B.3.2). Pretraining models with higher weight decay produces models with more structured representations, i.e., representations in which information is encoded in a more linearly accessible form. As a result, fine-tuning may effectively start at a better initialization and can focus on refining and aligning existing representations to the fine-tuning task rather than continuing to learn representations, leading to improved downstream performance. This hypothesis is consistent with previous findings that weight decay produces representations that are more transferable to downstream tasks in computer vision Lee et al. (2023). It is further supported by the observation that the linear separability of model representations (probing accuracy) is strongly positively correlated with downstream model performance (Appendix Figure 15).

### B.4.2 WEIGHT DECAY REDUCES THE RANK OF ATTENTION MATRICES

Previous work by Kobayashi et al. (2024) provides a theoretical argument that weight decay should reduce the rank of attention matrices. Recall that attention scores can be understood as a bilinear form $X^T W_{QK} X$ where $W_{QK} = W_K^T W_Q \in \mathbb{R}^{n_{embed} \times n_{embed}}$ is the product of the query and key matrices, and $X \in \mathbb{R}^{n_{embed} \times T}$ is the matrix of token embeddings (or hidden representations) for a sequence of length $T$. Now, the matrix $W_{QK}$ is naturally low-rank since its rank is at most $d_{head}$, which is usually significantly smaller than $n_{embed}$. Kobayashi et al. (2024) argue that weight decay should further reduce the rank of $W_{QK}$, as well as of the value-projection matrix $W_{VP} = W_P W_V \in \mathbb{R}^{n_{embed} \times n_{embed}}$. Concretely, they show that L2 regularization applied to the factored matrices $W_K$ and $W_Q$ becomes equivalent to nuclear norm regularization on their product $W_{QK}$, which is known to induce low rank by promoting sparsity in the singular values. While Kobayashi et al. (2024) also provide empirical evidence on the Pile, their experiments were relatively small-scale from today's perspective. We now revisit the impact of weight decay on the rank of attention in our more modern setup.

**Weight decay reduces the rank of attention, but default weight decay yields near full-rank matrices.** Figure 5 depicts the evolution of the pseudo-rank (Supplement F.2.1) of the attention matrices during the training of the OLMo-2-1B-20x models. From Figure 5, we observe that there is a monotonic relationship between the weight decay parameter and the rank of the attention matrices, where larger weight decay values reduce the rank of both $W_{QK}$ and $W_{VP}$. However, unlike what is observed in Kobayashi et al. (2024), we see that the default weight decay parameter of 0.1 yields near full-rank matrices. This observation is further confirmed by Figure 18, which shows that the attention matrices in the fully trained OLMo-2-1B model are nearly full-rank.

**Attention matrices are differentially affected by weight decay.** Another important observation from our experiments is that the rank of the matrix $W_{QK}$ seems to be significantly more sensitive to weight decay than $W_{VP}$. In our experiments, a weight decay of $\lambda = 1.0$ reduces the rank of $W_{QK}$ by roughly a factor of 2, which is a common rank reduction observed in the literature on low-rank matrices. In contrast, the matrix $W_{VP}$ is still close to full-rank even for a weight decay value of 1.0. These results are especially pronounced for Llama-2 models depicted in Figure 16, where the rank of $W_{VP}$ remains essentially stable up to a weight decay value of 1.0, after which the rank collapses—a transition that correlates with a significant drop in performance.

**Low-rank structure as a driver of adaptability.** The observation that increased weight decay leads to lower-rank attention matrices provides a potential explanation for why weight decay improves model plasticity. In machine learning literature, low-rank constraints are a canonical form of regularization that is often believed to encourage simpler, more robust hypotheses (Cai et al., 2010; Oymak et al., 2019; Hu et al., 2022). We conjecture that by encouraging $W_{QK}$ toward a lower-rank

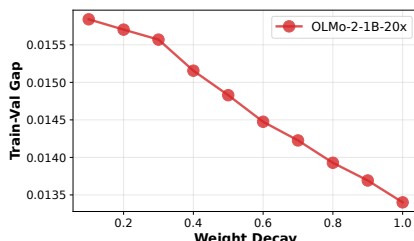

Figure 6: **Weight decay reduces overfitting on training data.** The figure depicts the train-val gap (Equation 3) for OLMo-2-1B models trained at 20 TPP.

configuration, weight decay may prevent the model from overfitting to high-dimensional noise in the pretraining distribution.

### B.4.3   WEIGHT DECAY REDUCES OVERFITTING ON TRAINING DATA

Lastly, we explore how weight decay influences the extent to which the pretrained model overfits the pretraining data. Previous work has shown that weight decay can cause the forgetting of individual benchmark questions seen during pretraining (Bordt et al., 2025). In the context of model plasticity, the ability to learn new information tends to be associated with the forgetting of prior data, a trade-off commonly referred to as the stability-plasticity dilemma Kirkpatrick et al. (2017); Riemer et al. (2018); Ibrahim et al. (2024); Elsayed & Mahmood (2024). Building on these insights, we investigate how weight decay influences overfitting, which is closely related to the forgetting of training data, in pretrained models.

To measure the degree to which a pretrained model overfits the training data, we compute the difference between the loss on the validation data and that on the training data:

$$\text{Train-Val Gap} = \text{Validation Loss} - \text{Training Loss} \tag{3}$$

Here, the training loss is the average loss that the fully trained model encounters on the training data, which is distinct from the training loss curve or the final training loss value. A model that does not overfit the training data would theoretically have a train-val gap of zero. In practice, a larger train-val gap indicates a higher degree of overfitting on the training data, thus less forgetting of the training data.

Figure 6 depicts the train-val gap for the OLMo-2 models trained at 20 TPP. We observe that the train-val gap decreases monotonically as the weight decay parameter is increased. This provides empirical evidence that models trained with larger weight decay values do indeed overfit the training data less.

**Reproducibility statement.** Upon publication of the paper, we will release code and models associated with the paper.

## C   PRE-TRAINING

### C.1   MODEL ARCHITECTURES AND TRAINING REGIMES

We pretrain models from different families (Llama-2 and OLMo-2), of different scales (up to 4B), and under different training regimes (20 TPP and 140 TPP), yielding five model setups. Details are in Table 1. For each model setup, we pretrain variants with varying weight decay values.

Table 1: **Model architectures.** We use Llama-2 model architectures from Qi et al. (2025) and OLMo-2 model architecture from OLMo et al. (2024). Llama-2 models are trained at 20 TPP and OLMo-2 models are trained at 20 TPP and 140 TPP.

|  | **Llama-2-0.5B** | **Llama-2-1B** | **Llama-2-4B** | **OLMo-2-1B** |
|---|---|---|---|---|
| Model size | 0.5B | 1B | 4B | 1.5B |
| Hidden size | 1536 | 2048 | 4096 | 2048 |
| Intermediate size | 3216 | 4896 | 7792 | 16384 |
| Vocab size | 32000 | 32000 | 32000 | 100278 |
| Context length | 2048 | 2048 | 2048 | 4096 |
| # Heads | 32 | 32 | 32 | 16 |
| # Layers | 20 | 22 | 28 | 16 |
| # Query groups | 4 | 4 | 4 | 16 |

## C.2 TRAINING DETAILS

The training data size (measured in tokens) for each model is determined by the TPP ratio.

Table 2: **Model configurations and training data sizes.**

| Model | TPP Ratio | Training Data Size |
|---|---|---|
| Llama-2-0.5B-20x | 20 | 10 BT |
| Llama-2-1B-20x | 20 | 20 BT |
| Llama-2-4B-20x | 20 | 80 BT |
| OLMo-2-1B-20x | 20 | 30 BT |
| OLMo-2-1B-140x | 140 | 210 BT |

To pre-train Llama-2 models, we use up to 8 A100 GPUs or 16 H100 GPUs. To pre-train OLMo-2 models, we use 8xH100 GPUs. The 20x OLMo-2-1B models are each trained for 2 days on a single H100 node. The 140x models are trained for 2 weeks on a single H100 node. For all models, we use the AdamW optimizer and standard warmup-cosine learning rate schedule. The only exception is the OLMo-2-1B 140x models, which follow a warmup-stable-decay schedule Hägele et al. (2024). OLMo-2-1B models are pretrained using the official AllenAI repository.

For each model, we train variants with various weight decay values specified in Table 3. Additional hyperparameters are in Table 4 and 5.

Table 3: **Weight decay values for each model.** For Llama-2-4B-20x, we use the weight decay 0.1 model from Qi et al. (2025) and pre-train the weight decay 1.0 model. For OLMo-2-1B-140x, we use the weight decay 0.1 model from Bordt & Pawelczyk (2025) and pretrain the weight decay 0.3 and 1.0 models.

| Model | Weight Decay |
|---|---|
| Llama-2-0.5B-20x | 9 values: {0.0001, 0.001, 0.01, 0.1, 0.5, 1.0, 1.5, 3.0, 10.0} |
| Llama-2-1B-20x | 9 values: {0.0001, 0.001, 0.01, 0.1, 0.5, 1.0, 1.5, 3.0, 10.0} |
| Llama-2-4B-20x | 2 values: {0.1, 1.0} |
| OLMo-2-1B-20x | 3 values: {0.1, 0.2, 0.3, 0.4, 0.5, 0.6, 0.7, 0.8, 0.9, 1.0} |
| OLMo-2-1B-140x | 3 values: {0.1, 0.3, 1.0} |

Table 4: **Hyperparameters for Llama-2 model pre-training.** For Llama-2 models, hyperparameter values are chosen following those in Qi et al. (2025), except for weight decay, which is varied as the independent variable in our experiments.

| Hyperparameter | Llama-2-0.5B-20x | Llama-2-1B-20x | Llama-2-4B-20x |
|---|---|---|---|
| precision | bf16-mixed | bf16-mixed | bf16-mixed |
| global_batch_size | 512 | 512 | 1024 |
| max_seq_length | 2048 | 2048 | 2048 |
| lr_warmup_ratio | 0.1 | 0.1 | 0.1 |
| max_norm | 1 | 1 | 1 |
| lr | 0.00025 | 0.0002 | 0.00015 |
| min_lr | 0.000025 | 0.00002 | 0.000015 |
| weight_decay | varies | varies | varies |
| beta1 | 0.9 | 0.9 | 0.9 |
| beta2 | 0.95 | 0.95 | 0.95 |
| epoch | 1 | 1 | 1 |

Table 5: **Hyperparameters for model pre-training.** For OLMo-2 models, hyperparameter values follow the OLMo-2 defaults (OLMo et al., 2024), except for weight decay, which is varied as the independent variable in our experiments.

| Llama-2-4B-20x | OLMo-2-1B-20x | OLMo-2-1B-140x |
|---|---|---|
| precision | bf16-mixed | bf16-mixed |
| global_batch_size | 512 | 512 |
| max_seq_length | 4096 | 4096 |
| lr_warmup_ratio | 0.1 | 0.1 |
| max_norm | 1 | 1 |
| lr | 0.0004 | 0.0004 |
| min_lr | 0.00004 | 0 |
| weight_decay | varies | varies |
| beta1 | 0.9 | 0.9 |
| beta2 | 0.95 | 0.95 |
| epoch | 1 | 1 |

# D FINE-TUNING

## D.1 FINE-TUNING DATASETS

We clean the fine-tuning training datasets, removing incoherent or questions that exceed the maximum input sequence length of the models. The size of the fine-tuning training set for each task and the test set used to subsequently evaluate model performance are shown in Table 6.

Table 6: **Fine-tuning and evaluation datasets.** MetaMathQA and SimpleScaling are evaluated on test sets of the GSM8KPlatinum Cobbe et al. (2021); Vendrow et al. (2025) and MATH Hendrycks et al. (2021) datasets because MetaMathQA and SimpleScaling contain questions that are augmented from the training sets of these two datasets.

| Task | Training set | Test set |
|------|-------------|----------|
| MetaMathQA | $n = 395,000$ | GSM8KPlatinum ($n = 1,209$) + MATH ($n = 5,000$) |
| MedMCQA | $n = 182,555$ | MedMCQA ($n = 4183$) |
| PubMedQA | $n = 211,168$ | PubMedQA ($n = 1000$) |
| MMLUProCoT | $n = 123,836$ | MMLUProCoT ($n = 567$) |
| RACE | $n = 92,737$ | RACE ($n = 4934$) |
| SimpleScaling | $n = 54,484$ | GSM8KPlatinum ($n = 1,209$) + MATH ($n = 5,000$) |

## D.2 TRAINING DETAILS

Table 7: **Hyperparameters for supervised fine-tuning.** We set batch_size = 64 due to computational constraints. We set n_epochs = 3 based on results from Qi et al. (2025) indicating that this setting leads to the best downstream performance. All other hyperparameters are from Qi et al. (2025).

|  | **1B and under** | **4B** |
|---|---|---|
| cutoff_len | 2048 | 2048 |
| batch_size | 64 | 64 |
| learning_rate | 0.00001 | 0.0000075 |
| lr_scheduler_type | cosine | cosine |
| warmup_ratio | 0.1 | 0.1 |
| n_epochs | 3 | 3 |

## D.3 TEMPLATE

We use the following template for supervised fine-tuning.

*Human: {question}*
*Assistant: {response}*

## E EVALUATION

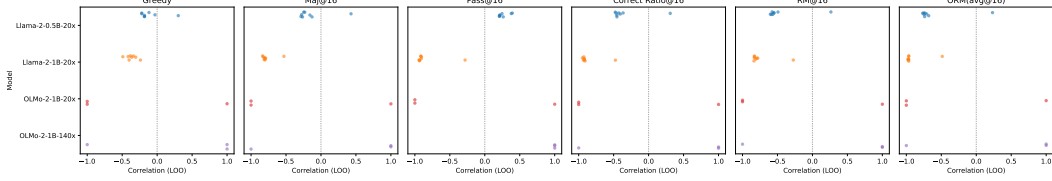

Figure 7: **Stability analysis for Pearson correlation coefficient.** Pearson correlation is computed for each leave-one-out (LOO) subset in Figure 9g. The LOO correlation can change noticeably in magnitude and sign, suggesting that the computed correlation relationship is rather unstable.

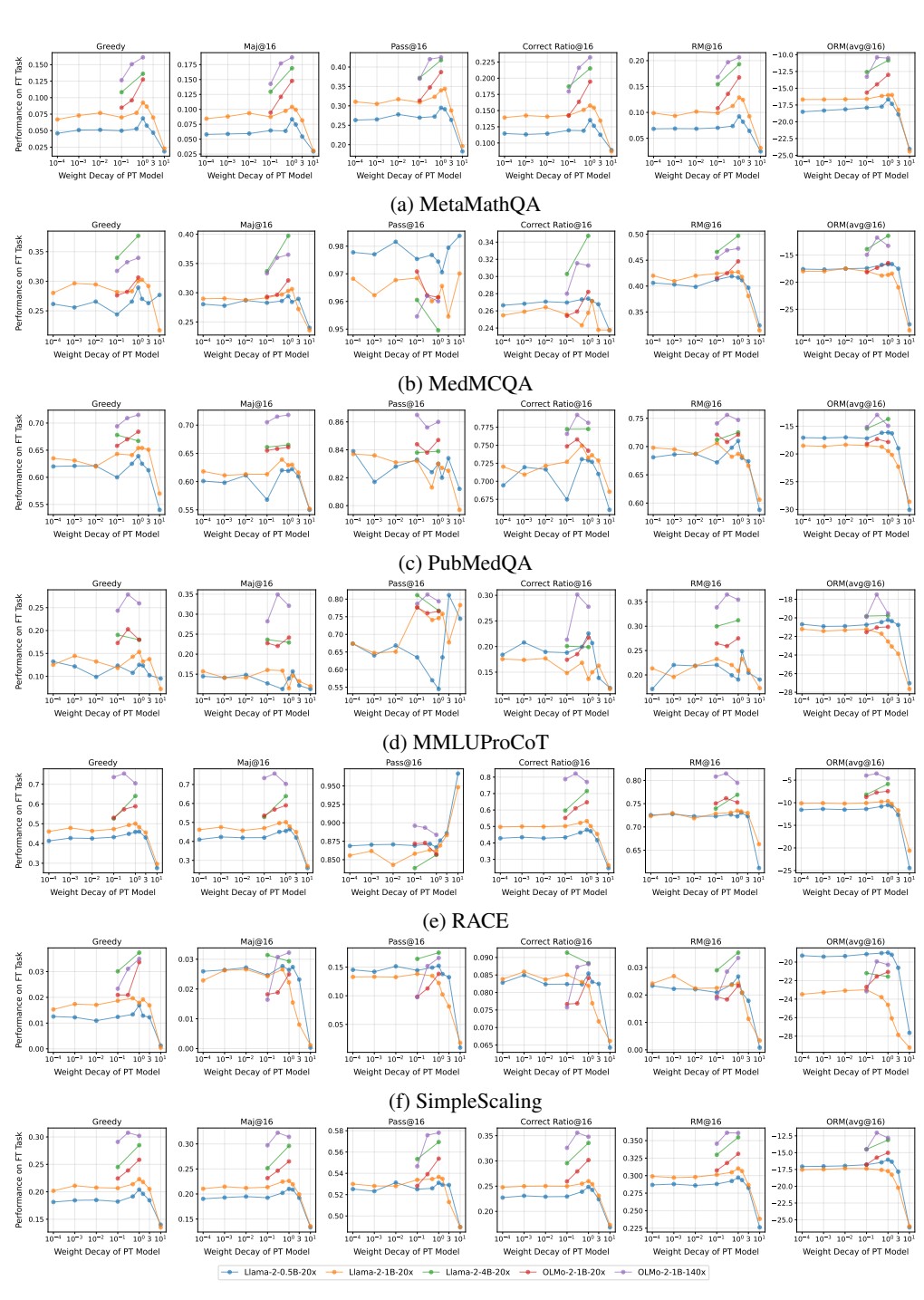

Figure 8: **Accuracy of models on each task after fine-tuning measured based on six metrics.**

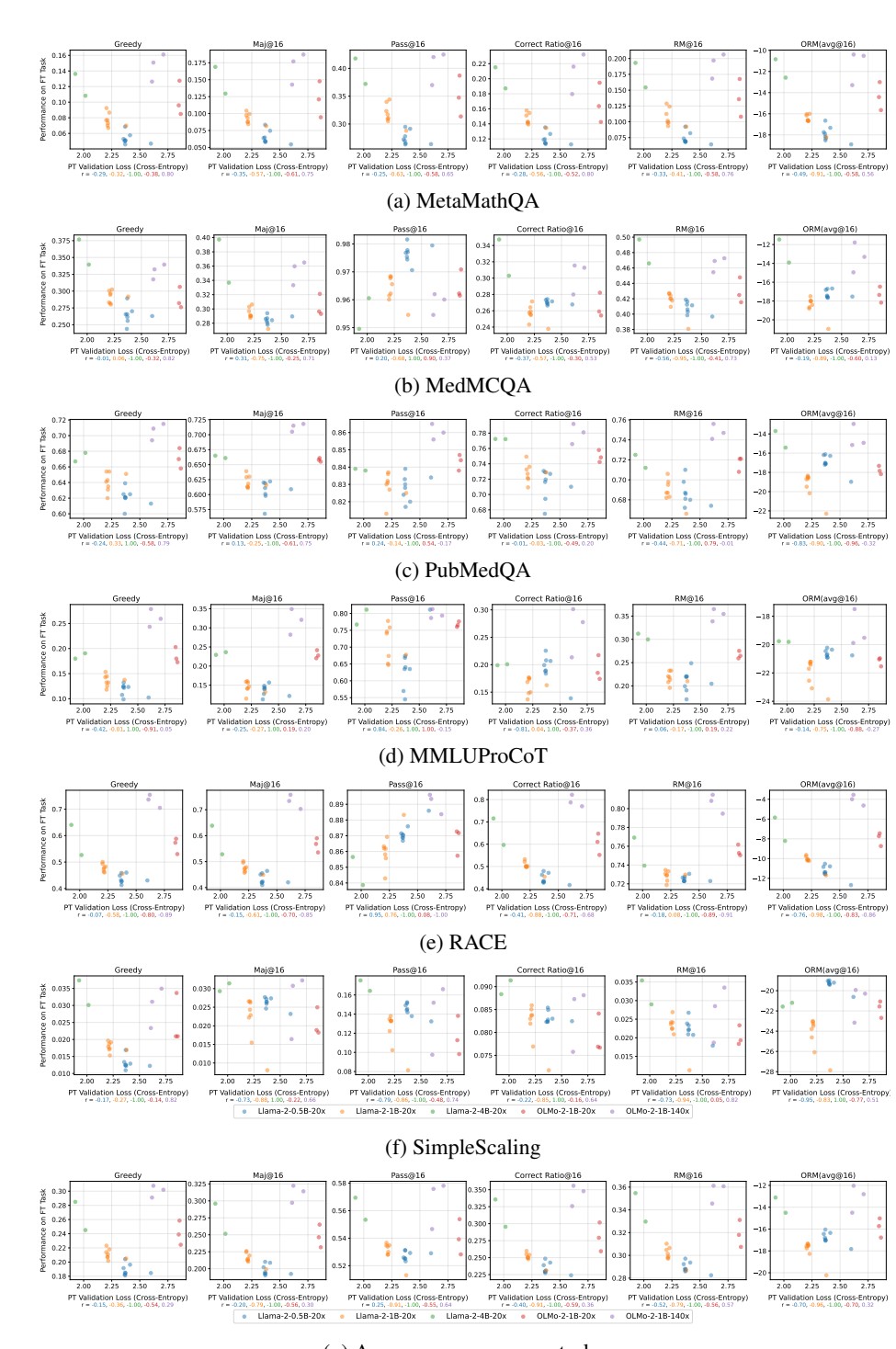

(a) MetaMathQA

(b) MedMCQA

(c) PubMedQA

(d) MMLUProCoT

(e) RACE

(f) SimpleScaling

(g) Average accuracy over tasks

Figure 9: **Loss after model pre-training is not predictive of model performance after fine-tuning.** Figures show model performance on individual datasets after fine-tuning measured based on six datasets.

# F ANALYSES ON WEIGHT DECAY'S MECHANISTIC EFFECTS ON MODEL BEHAVIOR

## F.1 WEIGHT DECAY'S EFFECT ON MODEL REPRESENTATIONS

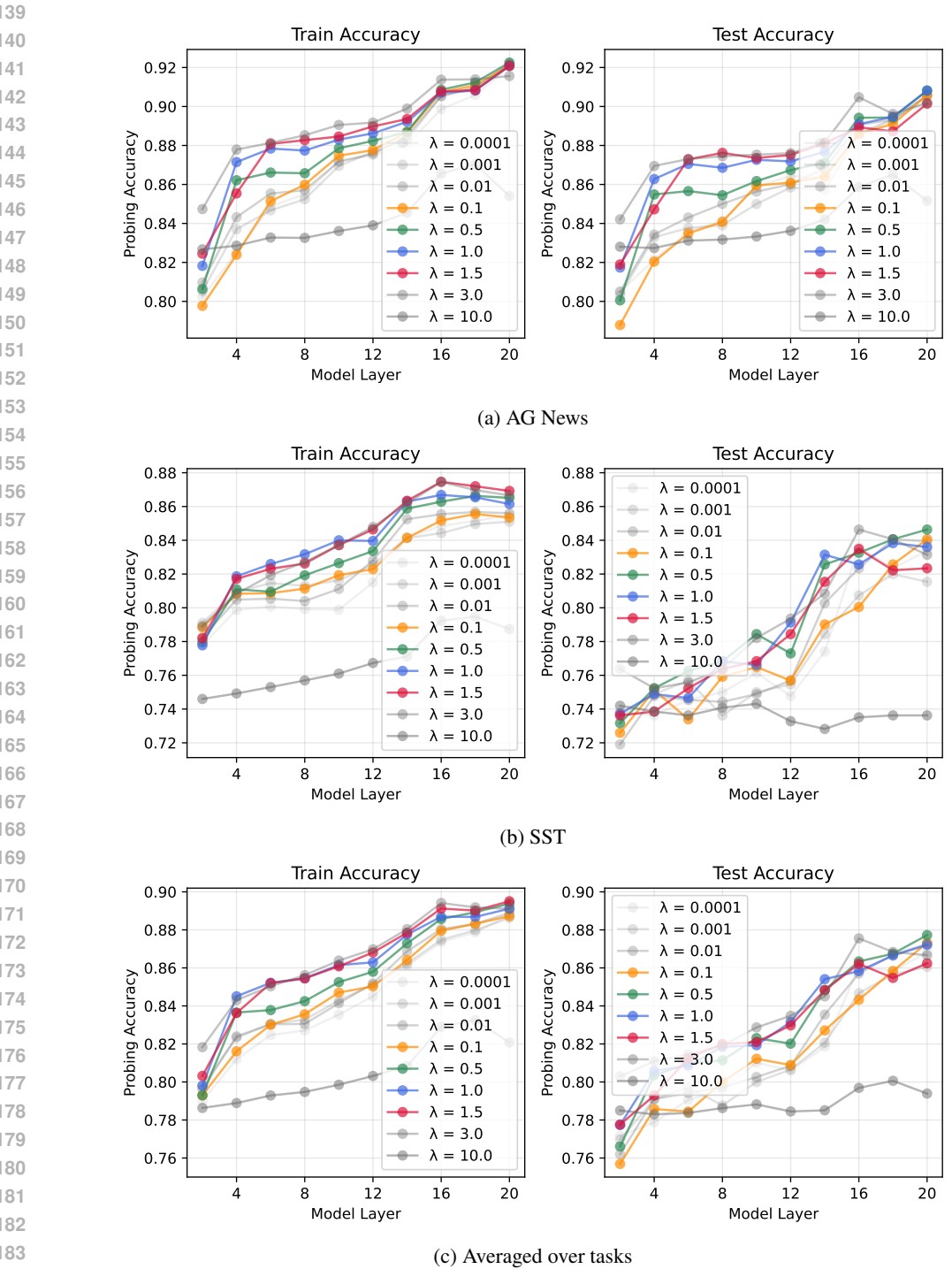

(a) AG News

(b) SST

(c) Averaged over tasks

Figure 10: **Linear probing experiments for Llama-2-0.5B-20x.** The train and test accuracies of the linear probes for the SST and AG News datasets and the average train and test accuracy over the two datasets.

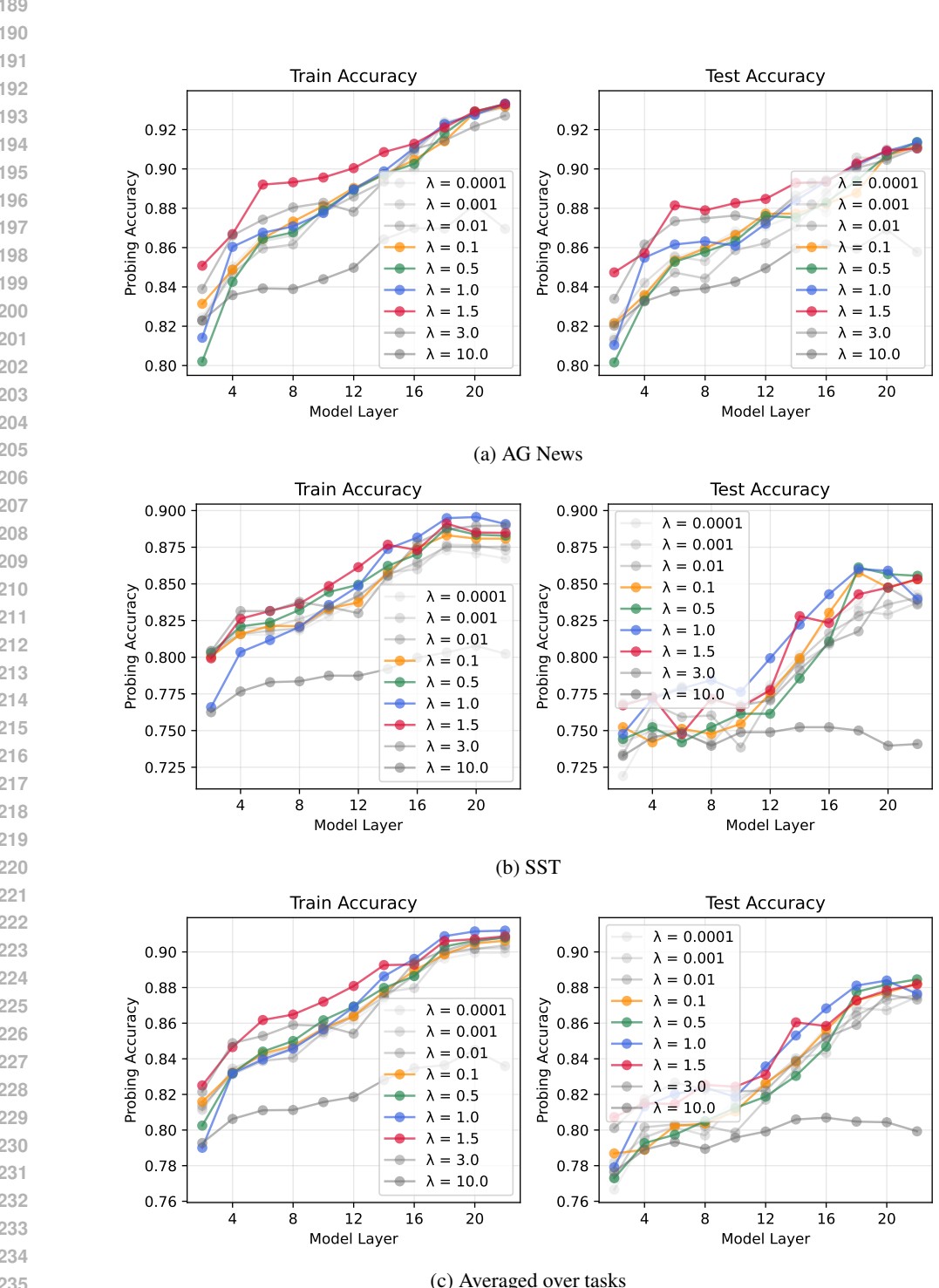

(a) AG News

(b) SST

(c) Averaged over tasks

Figure 11: **Linear probing experiments for Llama-2-1B-20x.** The train and test accuracies of the linear probes for the SST and AG News datasets and the average train and test accuracy over the two datasets.

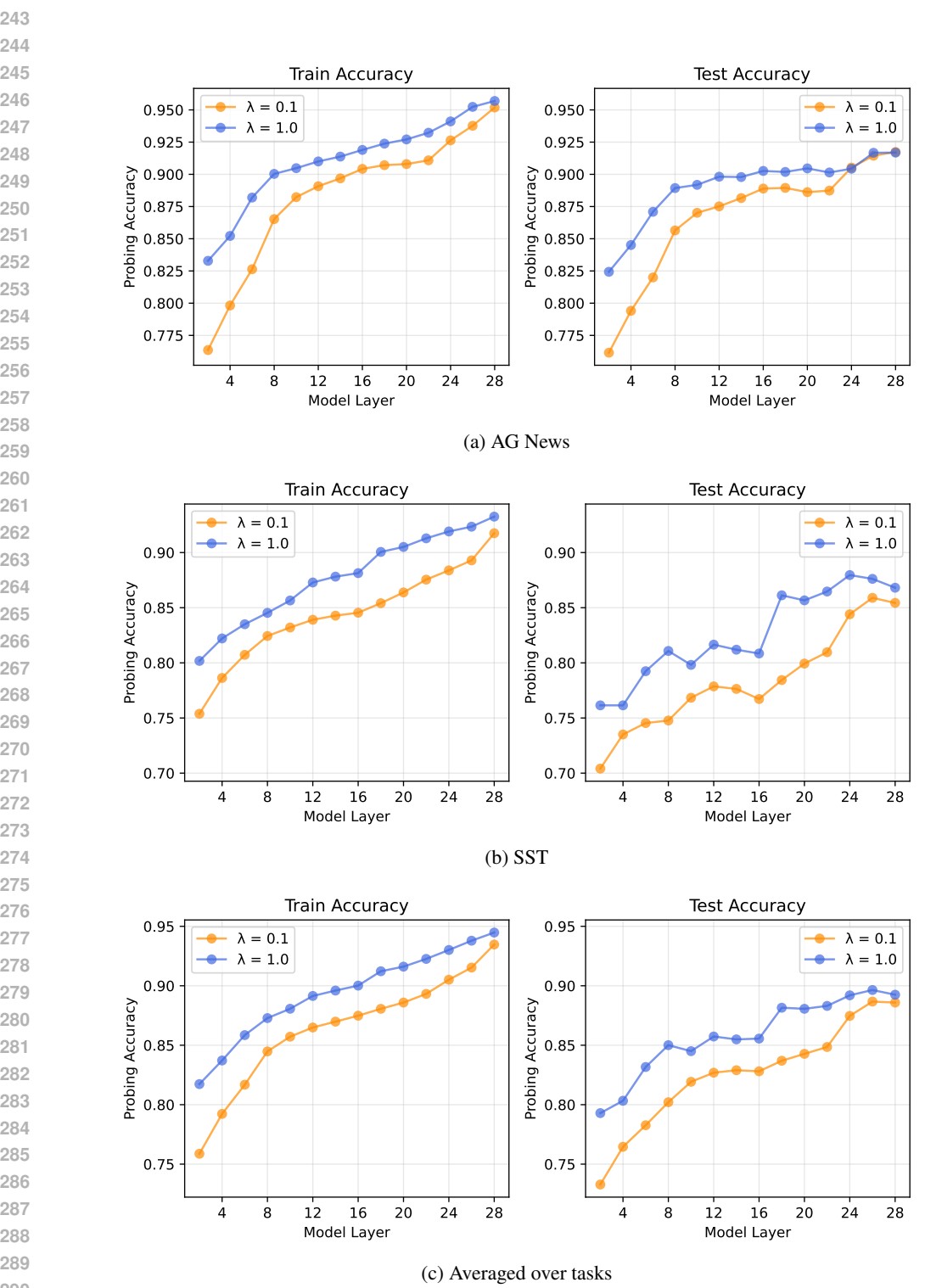

(a) AG News

(b) SST

(c) Averaged over tasks

Figure 12: **Linear probing experiments for Llama-2-4B-20x.** The train and test accuracies of the linear probes for the SST and AG News datasets and the average train and test accuracy over the two datasets.

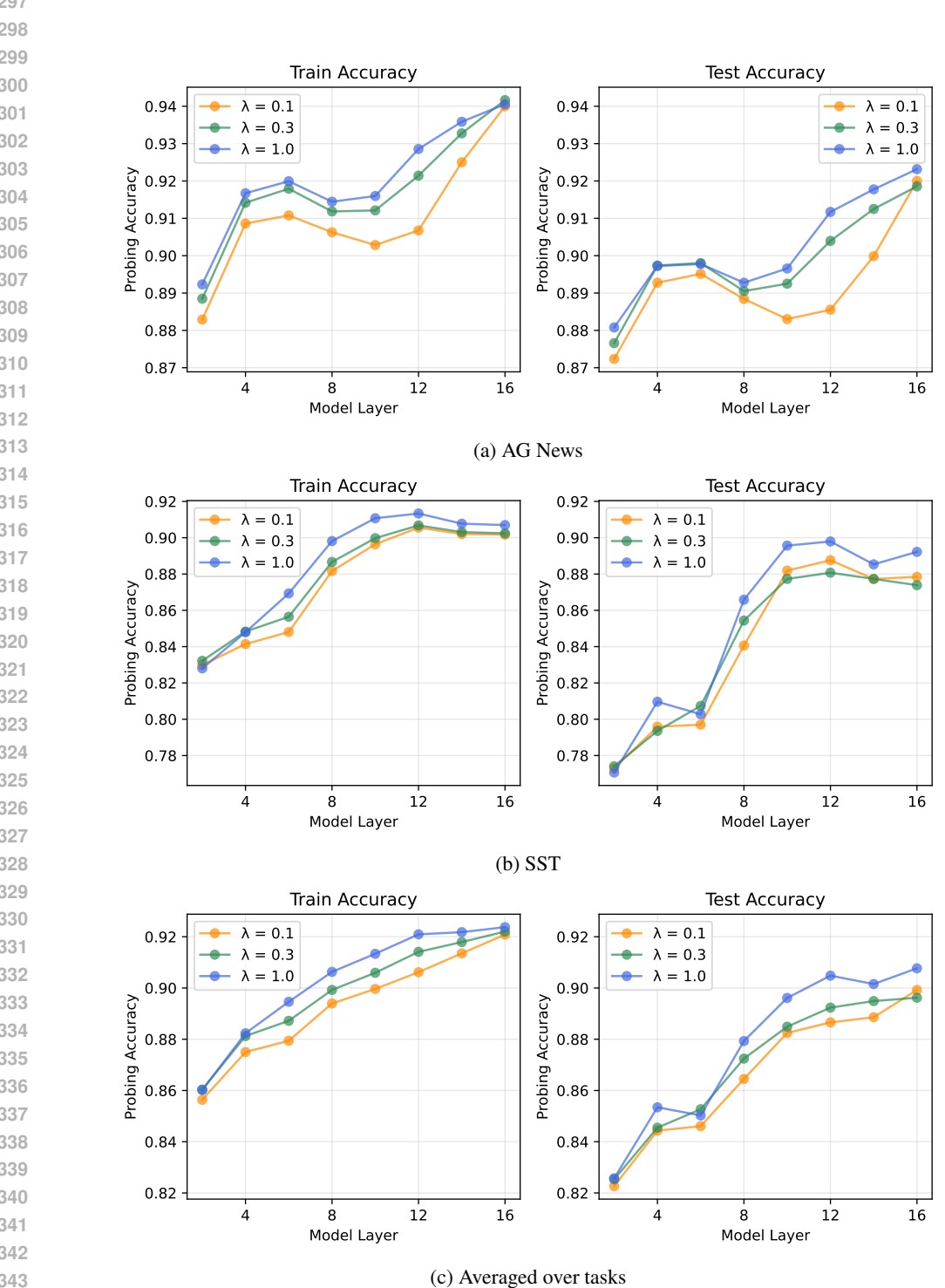

(a) AG News

(b) SST

(c) Averaged over tasks

Figure 13: **Linear probing experiments for OLMo-2-1B-20x.** The train and test accuracies of the linear probes for the SST and AG News datasets and the average train and test accuracy over the two datasets.

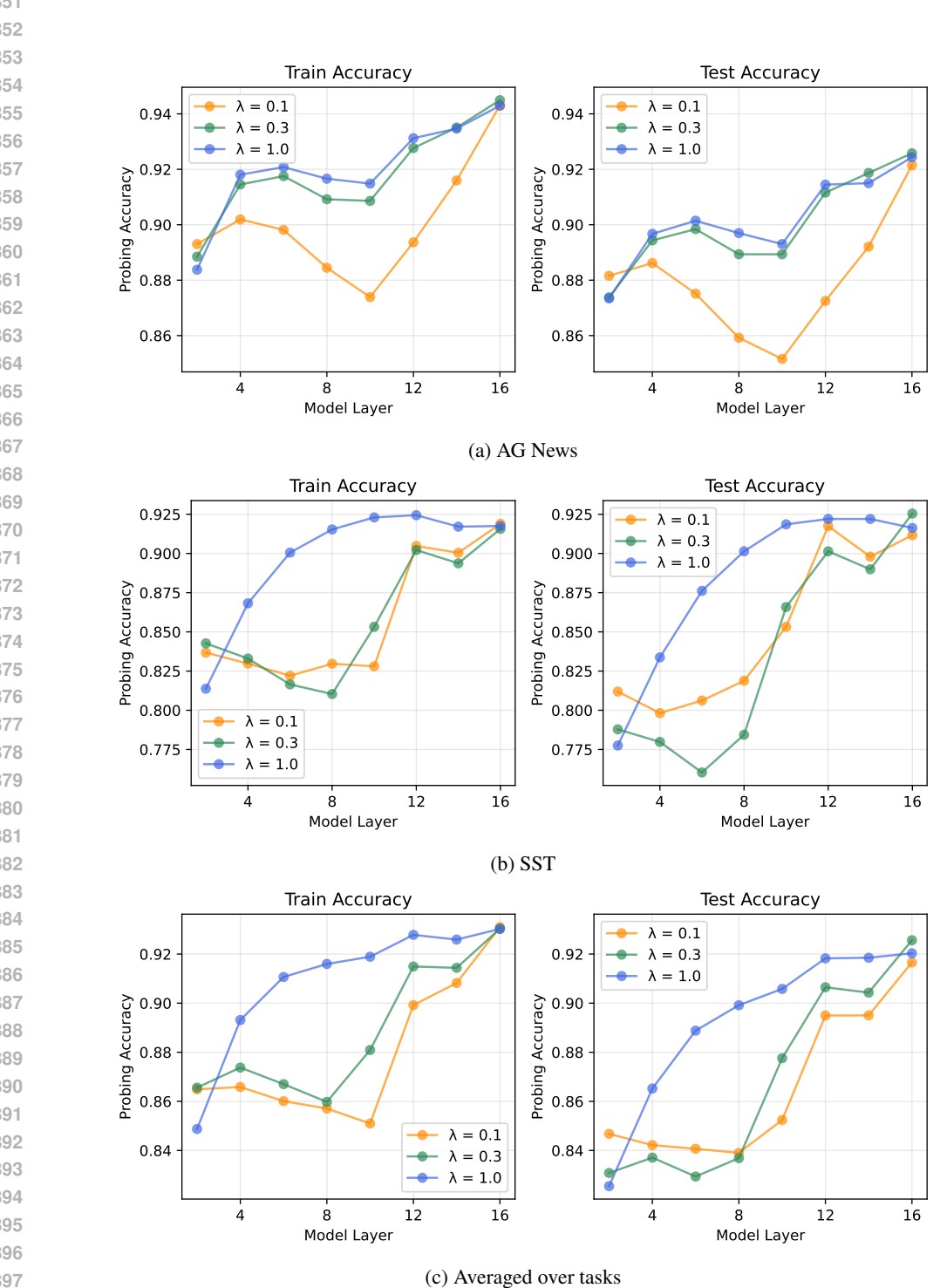

(a) AG News

(b) SST

(c) Averaged over tasks

Figure 14: **Linear probing experiments for OLMo-2-1B-140x.** The train and test accuracies of the linear probes for the SST and AG News datasets and the average train and test accuracy over the two datasets.

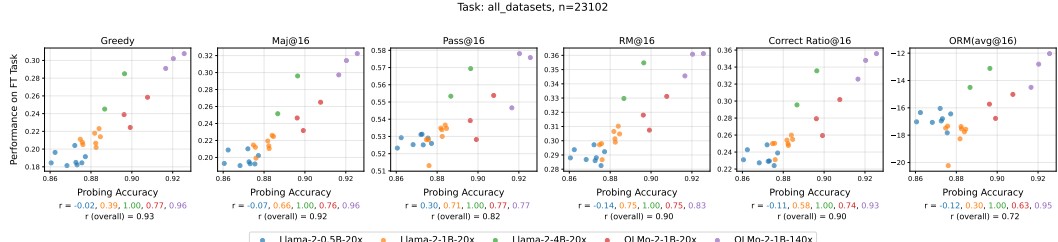

Figure 15: **Probing accuracy is highly predictive of downstream model performance.** The x-axis is the best average probing accuracy of the model (best out of all model layers). The y-axis the average accuracy of the model over all tasks after fine-tuning. Pretrained models with higher probing accuracies from the linear probing experiments tend to perform better downstream after fine-tuning.

### F.2 Weight decay's effect on attention matrix rank

#### F.2.1 Attention Pseudo-Rank Computation

To quantify the effective dimensionality of weight matrices, we follow Kobayashi et al. (2024) and compute the pseudo-rank of the matrices. For a matrix $W$ with singular values $\sigma_1 \geq \sigma_2 \geq \cdots \geq \sigma_n$, the pseudo-rank is defined as the ratio $k/n$, where $k$ is the smallest integer satisfying:

$$\frac{\sum_{i=1}^{k} \sigma_i}{\sum_{i=1}^{n} \sigma_i} \geq 0.95 \tag{4}$$

This metric represents the fraction of the largest singular values required to capture at least 95% of the sum of all singular values. In our analysis, we apply this computation to the product of the key-query matrices ($W_{QK} = W_K^T W_Q$) and the value-projection matrices ($W_{VP} = W_P W_V$) to monitor the emergence of low-rank structures during training.

#### F.2.2 Additional analyses on attention matrix rank

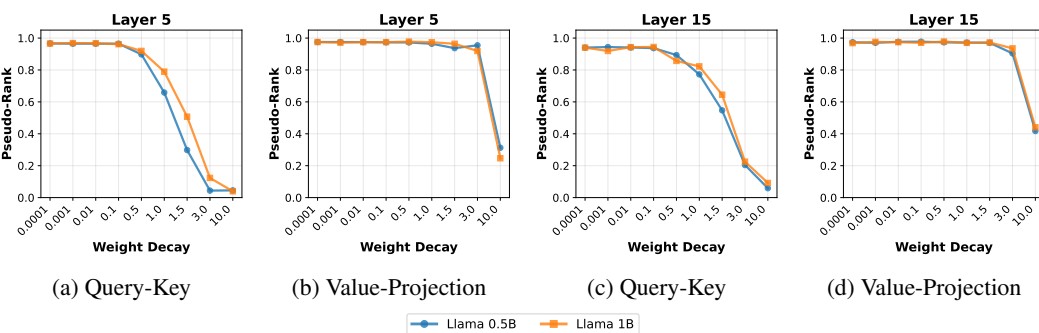

| (a) Query-Key | (b) Value-Projection | (c) Query-Key | (d) Value-Projection |

Figure 16: **Weight decay reduces the rank of attention matrices.** The figure depicts the average pseudo-rank (Supplement F.2.1) of the query-key ($W_{QK}$) and value projection ($W_{VP}$) matrices in layers 5 and 15 of the fully-trained Llama-2 models at 20 TPP.

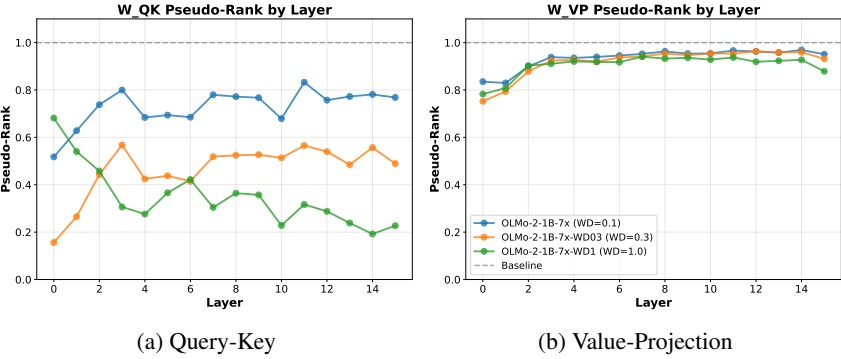

| (a) Query-Key | (b) Value-Projection |

Figure 17: **Weight decay reduces the rank of attention matrices.** This is for the OLMo models trained at 140 TPP. We observe that the rank of attention for weight decay 0.1 is generally smaller than what we observe for both for 20 TPP and for the fully trained OLMo-2-1B-0425 model. Hence, we conjecture that this is because the 140 TPP models were trained with a warmup-stable-decay learning rate schedule, whereas the 1x and 144x models were trained with a cosine learning rate schedule. While it has been shown that WSD leads to a similar validation loss to cosine decay (Hägele et al., 2024), there is emerging evidence that there are important differences between the training dynamics of the two learning rate schedules Catalan-Tatjer et al. (2025).

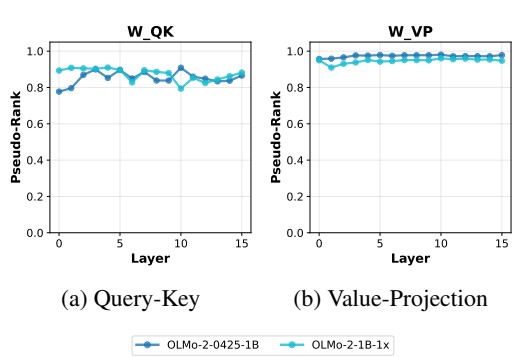

(a) Query-Key  (b) Value-Projection

Figure 18: **Training time does not reduce the rank of attention matrices.**

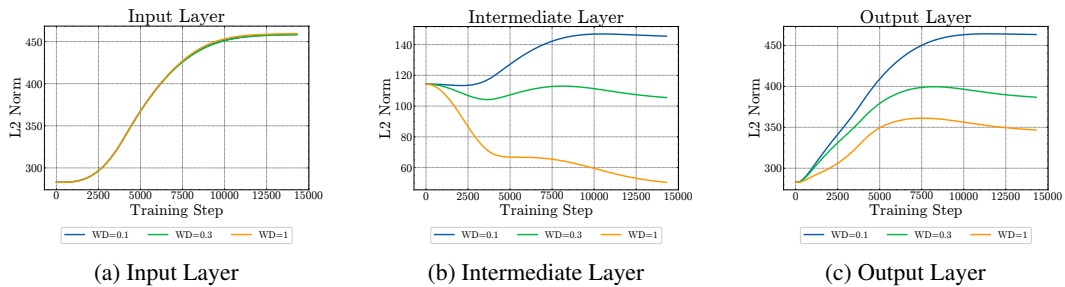

(a) Input Layer  (b) Intermediate Layer  (c) Output Layer

Figure 19: **Weight decay reduces the norm of the weights of the model.** The effect does not occur for the input layer, where the weights are not being decayed. This is for OLMo-2-1B models trained at 20 TPP.

