# OpenReview forum: "Weight Decay Improves Language Model Plasticity"
_ICLR.cc/2026/Workshop/Sci4DL — Sci4DL 2026_

### Official Review · Reviewer_oGwd · 2026-02-06

**Fit:** 2
**Significance:** 2
**Confidence:** 2

**Summary:**

This work study the role of weight decay for language model plasticity, namely the pretrained model's ability to adapt to downstream tasks during fine-tuning. The authors find that stronger weight decay improves plasticity, and provide mechanistic reasons from probing the learned representations, attention matrix ranks, and studying the overfitting phenomenon.

**Strengths:**

1. Understanding the trade-off between pre-training and fine-tuning is highly relevant to the community, and studying via the weight decay hyper-parameter is an interesting angle.
2. The authors provide mechanistic analyses (i.e., linear separated representation probing, attention matrix rank analysis) to study the effect of weight decay on the inner-workings of the trained model.

**Suggestions:**

1. Miss some related work, for example Liu, et al. "Same pre-training loss, better downstream: Implicit bias matters for language models." (ICLR 2023) which showed weight decay improves flatness, and flatness is correlated with downstream task performance.
2. It would be interesting to show a more fine-grained analysis of weight decay on different downstream desiderata, as the authors briefly commented as the end of the paper.

---

### Official Review · Reviewer_BPA1 · 2026-02-25

**Fit:** 3
**Significance:** 3
**Confidence:** 3

**Summary:**

This paper presents a large-scale empirical study investigating how the weight decay hyperparameter ($\lambda$) during the pretraining phase of Large Language Models (LLMs) affects their "plasticity"—that is, the model's capacity to adapt to new knowledge during subsequent supervised fine-tuning (SFT). The prevailing paradigm in LLM development heavily relies on optimizing pretraining hyperparameters to minimize the pretraining validation loss, under the widespread assumption that a lower pretraining loss guarantees better downstream performance. Through extensive experiments across multiple model families (Llama-2, OLMo-2), scales (up to 4B parameters), and training regimes (compute-optimal vs. overtrained), the authors reveal a counter-intuitive disconnect: the weight decay value that minimizes pretraining loss is frequently *not* the one that maximizes downstream SFT performance. Specifically, pretraining with a weight decay significantly larger than the default 0.1 (e.g., 0.5 or 1.0) often leads to a slightly higher pretraining loss but substantially better plasticity and downstream accuracy. Furthermore, the paper provides insightful mechanistic explanations for this phenomenon, demonstrating via linear probing and pseudo-rank analysis that larger weight decay encourages more linearly separable representations, induces low-rank attention matrices, and reduces overfitting on pretraining data.

**Strengths:**

*   **Profound Paradigm Shift (A Crucial Cautionary Tale):** While the paper primarily serves as an empirical unmasking rather than proposing a novel algorithmic architecture, this is precisely its greatest strength. By rigorously debunking the pervasive myth that "lower pretraining loss equates to better downstream fine-tuning," it exposes a critical blind spot in current LLM Scaling Laws and hyperparameter optimization practices.
*   **Rigorous and Extensive Experimental Validation:** The end-to-end experimental design is highly commendable. Testing across different architectures, scales, and critically, different training durations (20 TPP vs. 140 TPP) on multiple diverse Chain-of-Thought (CoT) tasks ensures the findings are robust and not merely artifacts of a specific setup.
*   **Deep Mechanistic Insights:** The transition from observing a macroscopic phenomenon (plasticity improvement) to explaining its microscopic drivers (white-box analysis) perfectly aligns with the mission of SciForDL. The use of pseudo-rank to show the collapse of $W_{QK}$ and linear probing to demonstrate structural representation changes provides a satisfying, scientifically grounded explanation for *why* weight decay enhances plasticity.
*   **Clarity and Presentation:** The paper is exceptionally well-structured and written. The figures (especially the disconnect shown in Figure 2 vs. Figure 3) effectively communicate complex empirical trade-offs.

**Suggestions:**

To further strengthen the scientific rigor of this excellent paper, the authors should address the following points regarding potential confounding variables and trade-offs:

**1. Disentangling the Learning Rate Schedule from Weight Decay Effects (Crucial Confounder):**
In Appendix F.2.2 (Figure 17), the authors note that the 140 TPP OLMo models utilized a Warmup-Stable-Decay (WSD) learning rate schedule, whereas the 20 TPP models used a standard cosine decay. The WSD schedule—particularly during its steep decay phase—can induce drastic representational shifts that might complexly interact with, or even overshadow, the effects of weight decay on plasticity.
*   *Suggestion:* The authors must explicitly acknowledge in the main text (e.g., in a Limitations or Discussion section) that the optimal weight decay shift observed between the 20 TPP and 140 TPP regimes might be partially confounded by the distinct LR schedules (Cosine vs. WSD). If computationally feasible, a small-scale ablation controlling for the LR schedule across different TPPs would vastly improve the rigor of this specific claim.

**2. The Trade-off Between Plasticity and Catastrophic Forgetting (General Capabilities):**
The paper successfully demonstrates that larger weight decay improves performance on specific fine-tuning CoT tasks (plasticity). However, high plasticity often comes at the cost of accelerated catastrophic forgetting of the broad knowledge acquired during pretraining.
*   *Suggestion:* If a model becomes highly "plastic" for CoT reasoning, does it simultaneously suffer severe degradation in general zero-shot capabilities? It would be highly valuable to evaluate the fine-tuned models on 1-2 standard general benchmarks (e.g., MMLU or PIQA) to see if the large weight decay models maintain their general knowledge better or worse than the default ($\lambda=0.1$) models. This would provide a holistic view of the stability-plasticity dilemma.

**3. Clarifying the "Tipping Point" of Rank Collapse:**
The mechanistic analysis of attention matrix rank (e.g., Figure 16) is excellent. However, there is a visible cliff where the rank of $W_{VP}$ collapses abruptly when $\lambda > 1.0$, which coincides with a sharp drop in performance.
*   *Suggestion:* The authors could briefly expand the discussion on this "sweet spot" of low-rank regularization. It appears weight decay must be large enough to enforce abstraction (low rank) but small enough to avoid total representational collapse. Explicitly linking the quantitative rank collapse point to the accuracy cliff would make the mechanistic explanation even more compelling.

---

### Meta-Review · Area_Chair_Z7qp · 2026-03-01

**Recommendation:** Accept

**Metareview:**

This work focuses on understanding weight decay for language model plasticity, ie pre-trained model's ability to adapt to downstream tasks during fine-tuning. It is a very good fit to the workshop and I recommend to accept the work.

---

### Decision · Program_Chairs · 2026-03-02

Accept